# Boosting hot electron flux and catalytic activity at metal–oxide interfaces of PtCo bimetallic nanoparticles

Hyosun Lee[1], Juhyung Lim[2], Changhwan Lee[1,2], Seoin Back[2], Kwangjin An [3], Jae Won Shin[1], Ryong Ryoo[1,4], Yousung Jung [2] & Jeong Young Park [1,2,4]

Despite numerous studies, the origin of the enhanced catalytic performance of bimetallic nanoparticles (NPs) remains elusive because of the ever-changing surface structures, compositions, and oxidation states of NPs under reaction conditions. An effective strategy for obtaining critical clues for the phenomenon is real-time quantitative detection of hot electrons induced by a chemical reaction on the catalysts. Here, we investigate hot electrons excited on PtCo bimetallic NPs during $H_2$ oxidation by measuring the chemicurrent on a catalytic nanodiode while changing the Pt composition of the NPs. We reveal that the presence of a CoO/Pt interface enables efficient transport of electrons and higher catalytic activity for PtCo NPs. These results are consistent with theoretical calculations suggesting that lower activation energy and higher exothermicity are required for the reaction at the CoO/Pt interface.

[1] Center for Nanomaterials and Chemical Reactions, Institute for Basic Science (IBS), Daejeon 34141, Republic of Korea. [2] Graduate School of EEWS, Korea Advanced Institute of Science and Technology (KAIST), Daejeon 34141, Republic of Korea. [3] School of Energy and Chemical Engineering, Ulsan National Institute of Science and Technology (UNIST), Ulsan 44919, Republic of Korea. [4] Department of Chemistry, Korea Advanced Institute of Science and Technology (KAIST), Daejeon 34141, Republic of Korea. Correspondence and requests for materials should be addressed to Y.J. (email: ysjn@kaist.ac.kr) or to J.Y.P. (email: jeongypark@kaist.ac.kr)

In the field of heterogeneous catalysis, there is much interest in understanding how hot electrons, which are associated with energy dissipation and conversion processes during surface reactions, affect catalytic activity and selectivity[1–3]. Extensive experimental studies demonstrate that hot electrons are crucial for explaining the kinetics of catalytic surface reactions because the transport of hot electrons facilitates the formation of a transient state in the molecules[4–6]. However, the microscopic mechanisms of hot-electron-mediated chemistry are still unclear because of the extremely fast relaxation of hot electrons via electron–electron and electron–phonon interactions[1].

Recently, metal–semiconductor catalytic nanodiodes have been developed as a powerful tool for detecting and utilizing hot electrons generated on nanocatalysts under various surface reactions[7–13]. The architecture of these devices allows for the quick extraction of hot electrons across the metal–semiconductor interface before thermalization, thereby providing key evidence of non-adiabatic charge transfer during surface reactions. In a previous study on Pt nanoparticle (NP)/Au/TiO$_2$ catalytic nanodiodes, we showed that the size-dependent catalytic activity of Pt NPs was quantitatively described by the chemicurrent, which is the flow of hot electrons generated on the Pt NPs during a chemical reaction (i.e. in this instance, H$_2$ oxidation)[9]. Furthermore, the kinetics of H$_2$ oxidation following the Langmuir–Hinshelwood mechanism were investigated by measuring the magnitude of the number of hot electrons at different concentrations of hydrogen using Pt NP/graphene/TiO$_2$ nanodiodes[10].

In contrast with conventional monometallic NPs, bimetallic NPs have opened a new pathway that could control the electronic structure and binding energy in catalysts, resulting in superior catalytic performance[14–16]. Despite considerable focus on various catalytic reaction studies (e.g., catalytic reforming reactions, pollution control, electrochemical catalysis)[17,18], there are still questions about the underlying causes of improved performance because the structure, chemical composition, and oxidation state of bimetallic NPs can change under reaction conditions[19–21]. Recently, the presence of oxide–metal interfacial sites formed by surface segregation of bimetallic NPs were specifically suggested to be responsible for increased catalytic activity[22,23]. However, the physical nature and fundamental role of oxide–metal interfaces are still elusive because of a lack of definitive evidence.

Herein, we report the real-time detection of hot electrons generated on bimetallic PtCo NPs during exothermic H$_2$ oxidation and clarify the origin of the synergistic catalytic activity of PtCo NPs with corresponding chemicurrent values. To investigate the dynamics of hot electrons on nanocatalysts, we use catalytic NP/Au/TiO$_2$ nanodiodes composed of stoichiometric PtCo bimetallic NPs prepared via the co-reduction method using two metals. In both chemicurrent and turnover rate measurements, we observe that the catalytic activity of the bimetallic PtCo NPs is significantly enhanced compared with monometallic Co or Pt NPs. Through X-ray photoelectron spectroscopy (XPS) analysis, transmission electron microscopy (TEM), and density-functional theory (DFT) calculations, we confirm that this improvement is attributed to the presence of a CoO/Pt interface stabilized on the PtCo NP surface under reaction conditions. By estimating the chemicurrent yield, we conclude that the catalytic properties of the bimetallic NPs are strongly governed by the oxide–metal interface, which facilitates hot electron transfer on the NPs.

## Results

### Catalytic nanodiodes with PtCo bimetallic nanoparticles.
We present a schematic diagram of a typical device (Fig. 1a) and its

energy band diagram (Fig. 1b), where catalytically active NPs (i.e., Pt, PtCo, Co NPs) were assembled as a two-dimensional (2D) array onto the active surface of Au/TiO$_2$ nanodiodes using the Langmuir–Blodgett technique. Here, the hot electrons generated on the NPs pass across the Au/TiO$_2$ Schottky interface if they obtain sufficient energy (1–3 eV) from the chemical reaction to transport irreversibly through the interface. Furthermore, because the thickness of the metal layer is less than the electron mean free path (<15 nm)[9], direct detection of the hot electrons is possible before thermalization caused by electron–electron scattering and electron–phonon coupling. To study the dynamics of hot electrons on NPs, monolayer arrays of NPs deposited on Au/TiO$_2$ nanodiodes were confirmed by scanning electron microscopy (SEM) (Fig. 1c–e and Supplementary Fig. 1). To estimate the conductive properties of the catalytic nanodiodes, current–voltage (I–V) curves were plotted (Fig. 1f). After fitting the I–V curves to the thermionic emission equation, we confirmed that a Schottky barrier was formed at the Au/TiO$_2$ interface with a height of 0.7 eV that was preserved regardless of the type of NPs (Fig. 1g; see Supplementary Note 1 for details). This indicates that the Schottky barrier height is not affected by the electronic properties of the NPs deposited on the Au/TiO$_2$ nanodiode.

### Structural and chemical characterization of the Pt$_x$Co$_y$ NPs.
We synthesized Pt$_x$Co$_y$ NPs with different compositions ($x{:}y =$ 1:0, 3:1, 1:1, 1:3, 0:1) using the co-reduction method with two precisely mixed metal precursors to study the dynamics of hot electrons on bimetallic NPs. The structures of the Pt$_x$Co$_y$ NPs were characterized by TEM, high-angle annular dark-field scanning transmission electron microscopy (HAADF-STEM), and STEM-energy dispersive X-ray spectroscopy (STEM-EDS). As shown in the typical TEM images and size distribution histograms (Fig. 2a–c and Supplementary Fig. 2), the as-synthesized Pt$_x$Co$_y$ NPs have a uniform size distribution, spherical shape, and high crystallinity. The PtCo bimetallic NPs (i.e., Pt$_3$Co$_1$, Pt$_1$Co$_1$, Pt$_1$Co$_3$) show an average size range of 2.5–2.8 nm, which are slightly larger than those of the monometallic Pt$_1$Co$_0$ NPs (1.7 nm) and smaller than those of the monometallic Pt$_0$Co$_1$ NPs (5.1 nm). The measured $d$-spacings of the PtCo bimetallic NPs are less than the value of the Pt (111) plane (2.26 Å). The elemental distributions of the synthesized NPs were assessed using STEM-EDS mapping, which demonstrates that the Pt and Co atoms were randomly mixed within the lattice by forming alloys with different stoichiometric ratios of Pt to Co (Supplementary Figs. 2 and 3).

Further chemical analysis of the Pt$_x$Co$_y$ NPs was carried out using both XPS and inductively coupled plasma mass spectroscopy (ICP-MS) in which the chemical compositions determined by both measurements were well matched with the targeted molar ratios of the Pt and Co precursors (Fig. 2d, Supplementary Fig. 4, and Table 1). By indexing X-ray diffraction (XRD) patterns (Fig. 2e), we examined the microstructure of the Pt$_x$Co$_y$ NPs and confirmed that the Pt-rich samples (Pt$_1$Co$_0$, Pt$_3$Co$_1$, and Pt$_1$Co$_1$) exhibit typical (111), (200), and (220) diffraction peaks of the crystalline face-centered cubic (fcc) Pt phase, respectively, with no observable impurities of CoO$_x$ or phase-segregated metals[16]. The peak positions were also shifted slightly to a higher angle as more Co was incorporated into the PtCo NP, thus demonstrating the formation of bimetallic NPs with a disordered alloy phase, which agrees with the STEM-EDS mapping. The lattice parameters calculated from the (220) diffraction peak are 3.925 and 3.851 Å for Pt$_1$Co$_0$ and Pt$_3$Co$_1$, respectively, which are consistent with the reported lattice constants of Pt and Pt$_3$Co$_1$[24,25]. However, the pure Co (Pt$_0$Co$_1$, green) NPs show the characteristic pattern indexed to the cubic Co$_3$O$_4$ structure[26], indicating that the metallic Co NPs are easily oxidized—even at atmospheric

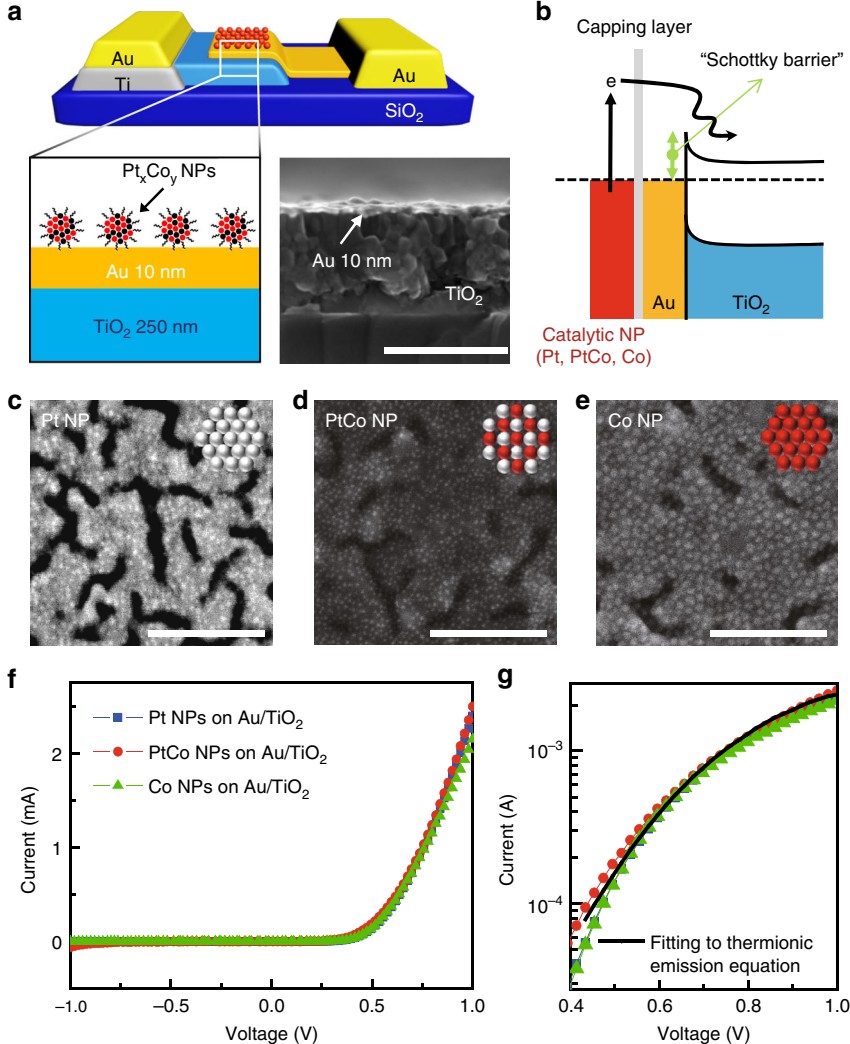

**Fig. 1** Detection of hot electrons generated on PtCo bimetallic NPs. **a** Schematic of a NP/Au/TiO$_2$ catalytic nanodiode and cross-sectional field emission scanning electron microscopy (FE-SEM) image of a 10 nm Au film on a 250 nm TiO$_2$ layer. The rectifying contact is formed at the interface between the Au and TiO$_2$ layers. The Ti layer makes the ohmic junction with the TiO$_2$. Scale bar in FE-SEM image is 300 nm. **b** Energy band diagram for the Au/TiO$_2$ nanodiode with various bimetallic NPs with different compositions. Hot electrons energetic enough to overcome the Schottky barrier can be detected as a steady-state current. Scanning electron microscopy (SEM) images of a monolayer of **c** Pt, **d** PtCo, and **e** Co NPs on a 10 nm Au layer. Scale bars are 100 nm (**c–e**). **f** Current–voltage (I–V) curves measured on the Au/TiO$_2$ catalytic nanodiodes with Pt (blue), PtCo (red), and Co (green) NPs. **g** Fitting the I–V curves of the Au/TiO$_2$ nanodiode to the thermionic emission equation. The catalytic nanodiodes show a Schottky barrier height of 0.7 eV

conditions—to Co$_3$O$_4$, which is thermodynamically more stable than the CoO phase. The average crystallite sizes were calculated from XRD using the Debye–Scherrer equation; they are consistent with the NP sizes obtained from TEM (Supplementary Table 2). Overall analyses confirm that the resulting Pt$_x$Co$_y$ NPs have uniform size and well-controlled stoichiometric Pt/Co ratios.

**Synergistic catalytic activity of the PtCo NPs**. As a model system, the catalytic oxidation of hydrogen (H$_2$ + O$_2$ → H$_2$O) has been chosen for the reaction kinetic study because it is a simple and representative reaction and it is closely correlated to a variety of hydrogen-based energy systems. To investigate the flow of hot electrons induced by the exothermic reaction on the catalytic NPs, we carried out current measurements on the Pt$_x$Co$_y$ NP/Au/TiO$_2$ catalytic nanodiodes in a H$_2$ (15 Torr) + O$_2$ (745 Torr) mixture as well as in pure O$_2$ at elevated temperatures (i.e., 30–120 °C). We observed a definite deviation between the

currents measured with and without catalytic H$_2$ oxidation, indicating that the differences in magnitude of the currents were associated with hot electrons (i.e., chemicurrent) generated on the surface of the Pt$_x$Co$_y$ NPs (Fig. 3a; see Supplementary Note 2 and Fig. 5 for details). The chemical nature of the observed currents was confirmed using a 30 nm Au layer where the chemicurrent drops to zero due to attenuation of the hot electrons[9].

Figure 3b shows the overall chemicurrent for the Pt$_x$Co$_y$ NPs; the current signals changed as a function of Co content and the Pt$_3$Co$_1$ bimetallic NPs exhibit the highest values during the catalytic H$_2$ oxidation reaction over the entire temperature range. In addition, it is noteworthy that the chemicurrents of the large NPs, including Pt$_3$Co$_1$, Pt$_1$Co$_1$, and Pt$_1$Co$_3$ (2.5 nm), exhibit a somewhat higher current than the smaller Pt$_1$Co$_0$ NPs (1.7 nm), even though the detection efficiency of hot electrons is higher in smaller NPs due to the shorter travel length for the hot electrons[9]. This implies that the enhanced chemicurrent obtained on the PtCo bimetallic NPs indeed originates from their improved catalytic properties and is beyond the size effect of the NPs.

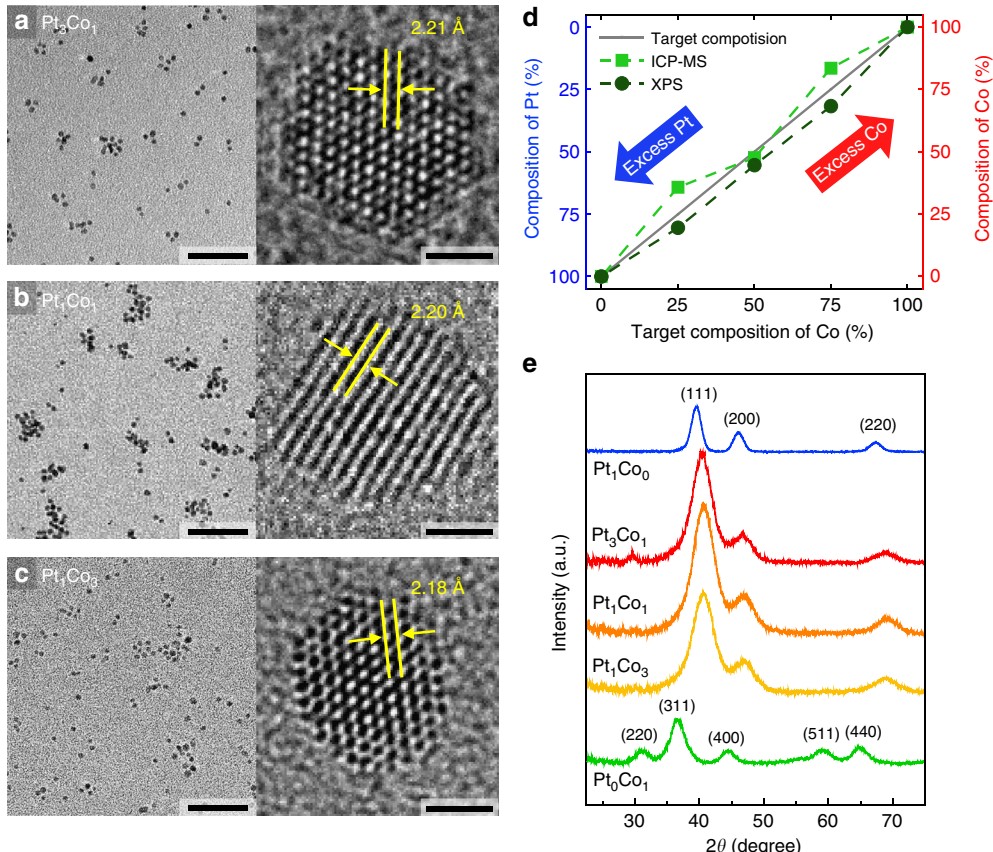

**Fig. 2** Structural and chemical characterization of PtCo bimetallic NPs. TEM and high-resolution TEM (HRTEM) images of as-synthesized **a** $Pt_3Co_1$, **b** $Pt_1Co_1$, and **c** $Pt_1Co_3$ bimetallic NPs. Scale bars are 30 nm (left) and 1 nm (right) (**a–c**). **d** Comparison of the composition values of the $Pt_xCo_y$ bimetallic NPs obtained from XPS and ICP-MS, which generally agree well with the targeted molar ratio of the Co and Pt precursors. **e** XRD patterns of $Pt_1Co_0$ (blue), $Pt_3Co_1$ (red), $Pt_1Co_1$ (orange), $Pt_1Co_3$ (yellow), and $Pt_0Co_1$ (green) NPs

Concurrently, the observed trend of the chemicurrent agrees well with the turnover frequency (TOF) measured on the mono-layered $Pt_xCo_y$ NPs supported on $SiO_2$ substrates at temperatures of 80–110 °C under identical reaction conditions (Fig. 3c). Here, to ensure the reversibility of the chemicurrent measurement, every measurement was repeated 2–3 times for all experiments and it was confirmed that the effect of the substrate on the catalytic activity is insignificant (Supplementary Figs. 7 and 8). The catalytic activities of the PtCo bimetallic NPs are significantly enhanced, while the monometallic $Pt_1Co_0$ or $Pt_0Co_1$ NPs show only moderate or no activity for the $H_2$ oxidation reaction, respectively. Moreover, as the Co content increased in the PtCo bimetallic NPs, both the chemicurrent and TOF values decreased and approached the signal of the $Pt_1Co_0$ NPs (Fig. 3d and Supplementary Fig. 9). These findings suggest that Co alloying with Pt at nanoscale has a synergistic effect on the catalytic reaction, and that the magnitude of the number of hot electrons captured by the catalytic nanodiodes is an instrumental descriptor of the catalytic properties regardless of the type of catalyst.

**Origin of the synergistic activity in the PtCo NPs**. Because of the complexity of the system, the origin of the synergistic catalytic effect found in bimetallic NPs is still debated. One of the most plausible reasons for the enhanced catalytic performance is the presence of oxide–metal interfacial sites, which are formed at the surface of the NPs during the catalytic reaction. To confirm the presence of CoO/Pt interfacial sites on the surface of our PtCo bimetallic NPs (i.e., $Pt_3Co_1$, $Pt_1Co_1$, $Pt_1Co_3$), we conducted ex situ

XPS and in situ TEM measurements. As shown in Fig. 4a, the chemical states of the cobalt in the $Pt_3Co_1$ NPs consist of metallic Co and CoO and the amount of CoO species increased by 23% because of further oxidation during $H_2$ oxidation. Unlike cobalt, regardless of the composition of the PtCo bimetallic NPs, platinum is rarely oxidized and maintains its metallic properties during the reaction (Supplementary Note 3 and Supplementary Figs. 10–14). Therefore, in the PtCo bimetallic NPs, it is definitive that the cobalt was easily oxidized and segregated on the metallic Pt surface as a form of CoO under the $H_2$ oxidation conditions[27,28]. We also noticed that the amount of CoO existing on the surface of the PtCo NPs increased as the Co/Pt ratio increased, resulting in a reduced interfacial area of the CoO/Pt on the NPs (Fig. 4b; see Supplementary Note 3 for more details).

Furthermore, to determine the structural configuration of the CoO layer formed in the PtCo NPs, in situ TEM experiments were performed using an aberration-corrected environmental TEM (Titan ETEM G2, FEI) with a Fusion heating holder (Protochips inc.), enabling dynamic observation of the NP surface at atomic-scale resolution (Supplementary Note 4). For the in situ TEM measurements, $Pt_3Co_1$ NPs were supported on spherical silica particles with diameters of 380 nm and the $Pt_3Co_1/SiO_2$ sample was heated from room temperature to 125 °C at a heating rate of 25 °C $min^{-1}$ in 0.5 mbar of $O_2$ gas inside the TEM. While monitoring the change in the surface structure of the $Pt_3Co_1$ NPs under oxidation conditions (0.5 mbar $O_2$ at 125 °C), we found that additional Co atoms began to segregate after a few seconds (~14 s), grew atom-by-atom along the {111} surface of the $Pt_3Co_1$ NP (~54 s), and formed a monolayer of CoO {111}, which is

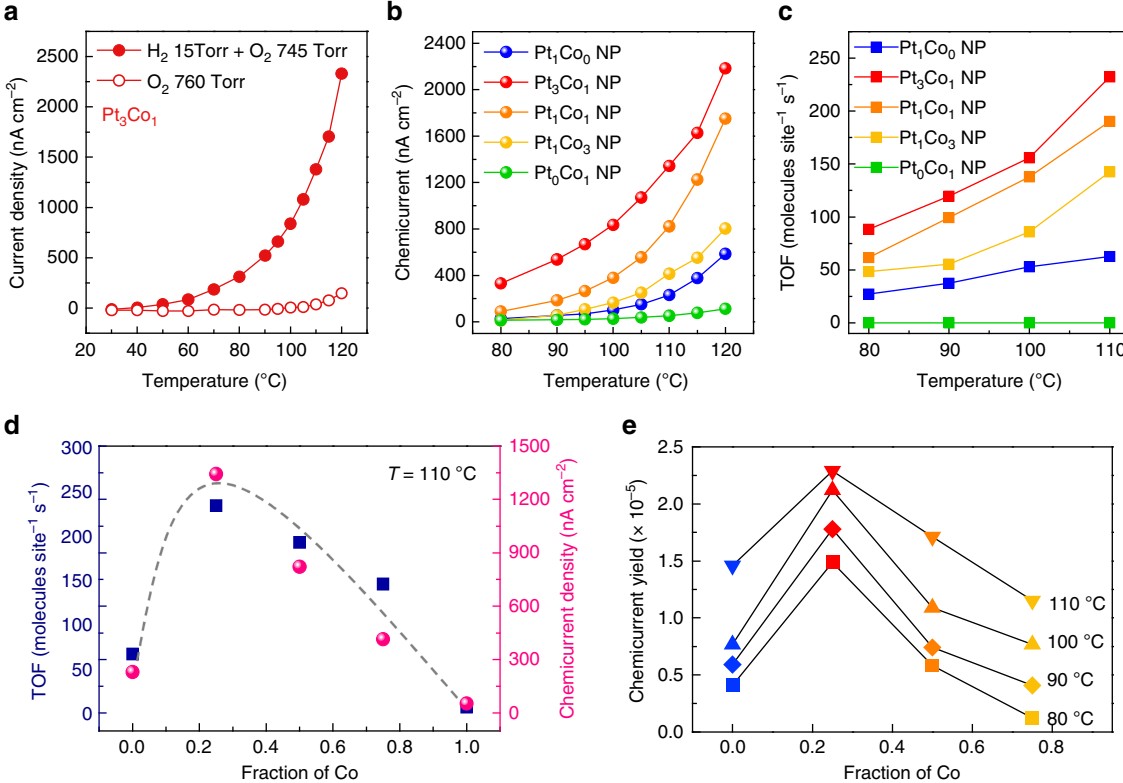

**Fig. 3** Hot electron detection and catalytic activity on PtCo bimetallic NPs. **a** Temperature dependence of the current from the Au/TiO$_2$ nanodiodes with Pt$_3$Co$_1$ NPs measured in the H$_2$ + O$_2$ gas mixture and in pure O$_2$. **b** Chemicurrents associated with the H$_2$ oxidation reaction measured on the Au/TiO$_2$ nanodiodes at different temperatures with Pt$_x$Co$_y$ NPs of different compositions; the data were normalized based on the total NP surface area. **c** Catalytic activity (i.e., TOF) for H$_2$ oxidation on a set of Pt$_x$Co$_y$ NPs with different compositions. **d** Comparison of both the chemicurrent and TOF as a function of the composition of the PtCo bimetallic NPs at 110 °C. **e** Plot of the chemicurrent yield as a function of the fraction of Co during H$_2$ oxidation at different temperatures

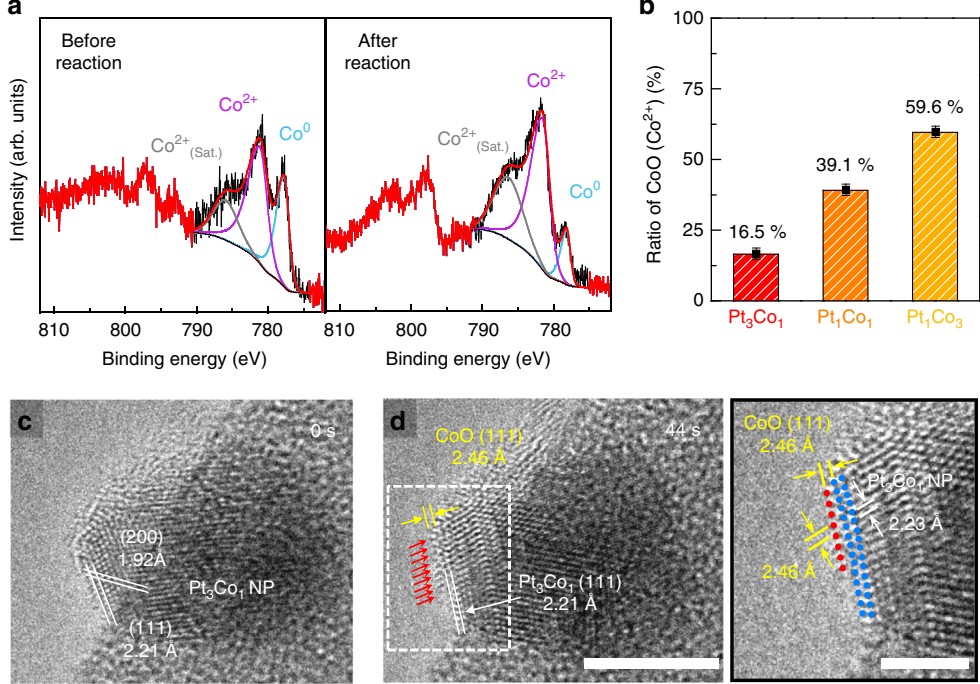

**Fig. 4** Formation of CoO on PtCo bimetallic NPs. **a** Co 2p XPS spectra showing the oxidation states of the Co in the Pt$_x$Co$_y$ bimetallic NPs before and after H$_2$ oxidation. **b** Relative ratio of CoO in the Pt$_x$Co$_y$ bimetallic NPs estimated using the peak area of Co$^{2+}$. Error bars are ± sd. Sequential in situ TEM images of the Pt$_3$Co$_1$ NP taken at **c** 0 and **d** 36 s during oxidization at 125 °C in 0.5 mbar of O$_2$ gas. As denoted by the red arrows, additional Co atoms segregated on the Pt$_3$Co$_1$ NP surface. The difference in lattice spacing between the top-most layer and the inner portion is clearly shown in the enlargement region denoted by the white rectangle. Scale bars are 5 nm (left) and 2 nm (right)

identified by lattice spacing measurements (Fig. 4c, d, Supplementary Fig. 15 and Movie 1). The red arrows on the top surface in Fig. 4d indicate Co atoms segregated on the NP surface. The lattice spacing between the outmost surface and the subsurface layer is 2.46 Å, which is larger than the 2.21 Å of $Pt_3Co_1\{111\}$ and identical to the d-spacing of CoO {111} (Fig. 4d). It was also confirmed that the CoO formed on the top-most layer of the NP was maintained even after 15 min (Supplementary Fig. 16). The formation and stabilization of the CoO layer on the PtCo NPs under oxidizing conditions is plausible because of the dominant interfacial strain at the CoO/Pt caused from lattice mismatch, or the strong interaction (i.e., so-called interface confinement effect) of Co–Pt binding at the CoO/Pt surface[29–31]. Therefore, the much-improved catalytic activity of the PtCo bimetallic NPs could be ascribed to the presence of CoO/Pt interfacial sites on the catalyst surface; and consequently, increasing the interfacial region between the Co and Pt on the PtCo NPs leads to a higher catalytic activity for $H_2$ oxidation.

However, the unique electronic properties of the oxide–metal interface are still somewhat ambiguous, especially at nanoscale, due to a lack of concrete experimental data[32]. To overcome this challenge, we estimated the chemicurrent yield, which is the detection probability of hot electrons when one molecule of product is formed during the surface reaction (Fig. 3e). Because the magnitude of the chemicurrent is proportional to the reaction rate, the chemicurrent can be expressed as $I_{ch} = \alpha q A N \cdot TOF$, where $\alpha$ is the chemicurrent yield, $q$ is the elementary charge, $A$ is the active area of the catalyst, and $N$ is the number of metal sites per square millimeter[1]. Here, the chemicurrent yield is determined by the distribution of hot electrons, attenuation in the metal, and the transmission probability across the Schottky barrier. Therefore, if both the NP size and Schottky barrier height of the nanodiodes remain the same, the chemicurrent yields are consistent at a given temperature. However, as shown in Fig. 3e, the chemicurrent yield varies depending on the composition of the PtCo NPs at a constant temperature, and the highest value is measured on the $Pt_3Co_1$ NPs with the largest interfacial area of CoO/Pt.

This intriguing result could be attributed to local polarization at the CoO/Pt interface that originates from charge transfer between the metal and the oxide, which contributes to an acceleration of

hot electron transport and to prolonging the lifetime of the hot electrons at the interfacial sites[32,33]. Therefore, the decrease in chemicurrent yield with increased CoO coverage indicates that the interfacial CoO/Pt region is reduced, resulting in less-efficient electron transfer on the PtCo NPs. These are important findings because they are the first visualization of the unique electronic structure of the oxide–metal interface in bimetallic NPs during a chemical reaction obtained through direct measurement of hot electrons.

## Discussion

To gain insight into the enhanced catalytic activity of the interfacial sites between the CoO and Pt, we investigated the reaction energetics of the $H_2$ oxidation reaction using density-functional theory (DFT) calculations[34]. For our model system, the CoO island structure on a Pt (111) surface, denoted as CoO/Pt, was investigated and compared with the Pt (111) surface (Supplementary Fig. 17)[29,35]. *OH formation is calculated to be the rate-determining step in the $H_2$ oxidation reaction for both the Pt (111) and CoO/Pt interface sites (Fig. 5a, b)[36–38]. However, the activation barrier for *OH formation at the CoO/Pt interface is smaller than that for Pt(111) by 0.16 eV. This result can be understood by the Pt–O bond at the CoO/Pt interface not undergoing major atomic rearrangement when forming *OH, while the same Pt–O bond on the Pt (111) surface requires the transfer of *O from the hollow site to the top site to form *OH (Supplementary Fig. 18 and Table 3). In the subsequent reaction steps on the CoO/Pt interface, *H migrates from the CoO to the nearby *O site on the Pt, followed by *H–*OH coupling to release $H_2O$ (Fig. 5c; see Supplementary Note 5 for more details). We also confirmed that the energy released during $H_2$ oxidation at the CoO/Pt interface (1.17 eV) is higher than that on the Pt (111) surface (0.79 eV), supporting the experimental results that hot electron generation is enhanced on the PtCo bimetallic NPs with a CoO/Pt interface. All these results thus point to the conclusion that local electronic perturbations caused by the presence of the CoO/Pt interface do indeed contribute to enhanced catalytic activity of the PtCo bimetallic NPs. Since it is possible that the reaction could also occur on the CoO cluster (island) itself instead

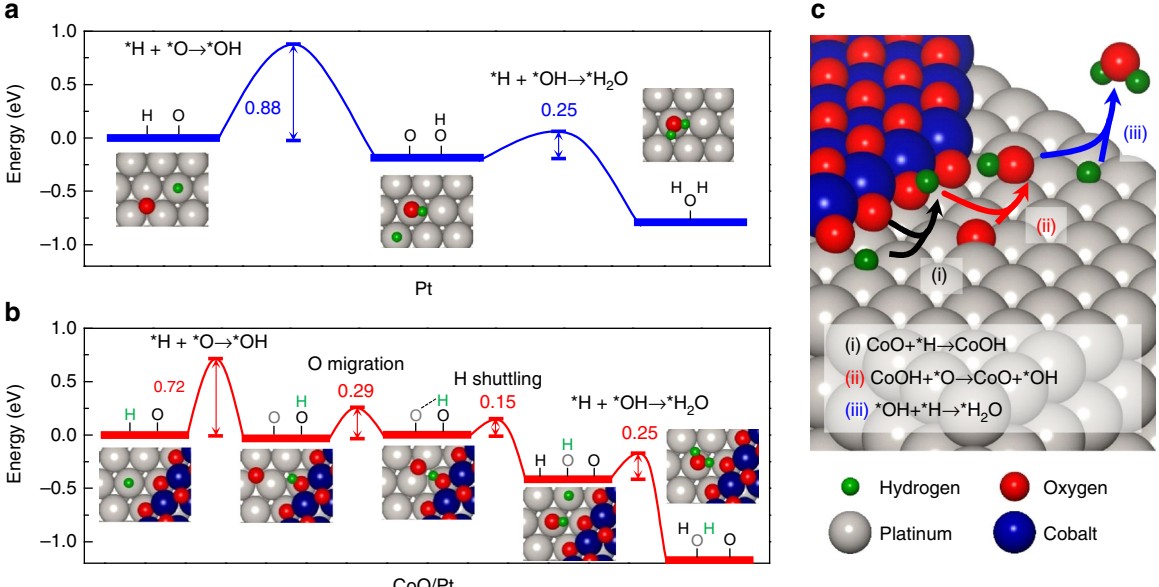

**Fig. 5** Reaction pathways of $H_2$ oxidation calculated using DFT. Schematic energy diagram of the $H_2$ oxidation reaction **a** on the Pt (111) surface and **b** at the CoO/Pt interface. **c** Drawing showing the $H_2$ oxidation reaction mechanism at the periphery of the CoO/Pt interface

of at the CoO/Pt interface, we also considered the same reactions occurring at the top sites of the island model of the cobalt monoxide. The high activation barrier (1.34 eV) indicates that $H_2$ oxidation on the cobalt monoxide cluster itself will be unlikely (Supplementary Fig. 20).

In this study, we have quantitatively correlated the catalytic activity of PtCo NPs with the magnitude of hot electron flow generated during $H_2$ oxidation via real-time chemicurrent measurements using a metal–semiconductor catalytic nanodiode. Based on both the chemicurrent and TOF results, we conclude that the synergistic catalytic performance of the PtCo NPs originates from the presence of the CoO/Pt interface under reaction conditions. Using DFT calculations, we also confirmed that the reduced energy barrier for OH formation at the CoO/Pt interface promoted the overall catalytic $H_2$ oxidation reaction. Most importantly, the composition-dependent chemicurrent yield proved that the locally modified electronic structure at the metal–oxide interface played a decisive role in improving the catalytic activity of the PtCo NPs. The present measurements are the first visualization of the unique electronic structure of the oxide–metal interface in bimetallic NPs during a chemical reaction, which were obtained through direct detection of hot electrons. This provides a clear message that chemicurrent experiments are a reliable way to investigate hot electron dynamics on nanocatalysts during chemical reactions.

## Methods

**Synthesis of the $Pt_xCo_y$ NPs.** To prepare the PtCo bimetallic NPs with well-controlled stoichiometry, we used the conventional polyol reduction method in which co-reduction of two metal precursors occurred and the initial concentration ratio of the cobalt-to-platinum precursors determined the final stoichiometry. Further details of the synthesis are described in the Supplementary Methods.

**Fabrication of the catalytic nanodiode.** The metal−semiconductor nanodiodes were fabricated as follows: First, we deposited a thin (250 nm) layer of Ti on a 500 nm $SiO_2$ layer/p-type Si (100) using electron beam evaporation with a titanium target and a patterned aluminum shadow mask ($4 \times 6$ mm$^2$). To modify the Fermi level of the $TiO_2$ film, the wafer was annealed in air at 380 °C for 2 h, while the sheet resistance was monitored. The electrode, a 50 nm Ti layer and 150 nm Au layer, was then deposited using electron beam evaporation through a second aluminum shadow mask ($5 \times 5$ mm$^2$). The titanium layer is the ohmic contact with the $TiO_2$ thin film and the Au layer comprises the nanodiode's two ohmic electrodes. Finally, a 10 or 30 nm Au thin film was deposited through a third patterned mask ($2 \times 1$ mm$^2$). This produced Au/$TiO_2$ nanodiodes with a 4 mm$^2$ active area.

To assemble the 2D monolayer of $Pt_xCo_y$ NPs (x:y = 1:0, 3:1, 1:1, 1:3, 0:1) on the Au/$TiO_2$ nanodiode, we used the Langmuir–Blodgett technique. The solution of $Pt_xCo_y$ NPs was first dispersed in water on a Langmuir–Blodgett trough (Nima Technology, M611) at room temperature. As the water-supported thin film layer of nanoparticles reached equilibrium, the layer was compressed by moving the mobile barrier at a rate of 15 cm$^2$ min$^{-1}$ while monitoring the surface pressure with a Wilhelmy plate. Finally, 2D arrays of $Pt_xCo_y$ NPs on the Au/$TiO_2$ nanodiode were created by lifting the submerged substrate from the water. Before setting the diode into the batch reactor, the nonactive portion of the diode surface was covered with Teflon tape, and the $Pt_xCo_y$ NPs were left only on the active diode area (i.e., the thin 10 nm Au layer in contact with the $TiO_2$) after removing the Teflon tape. The resulting monolayer arrays of NPs on the Au/$TiO_2$ nanodiode and $SiO_2$ substrate were confirmed by SEM, as shown in Fig. 1 and Supplementary Fig. 1, respectively.

**TOF measurement.** The $H_2$ oxidation reaction was performed in an ultra-high vacuum batch reactor (1 L) with a base pressure of $5.0 \times 10^{-8}$ Torr. The reaction chamber was evacuated and isolated with a gate valve before it was charged with 15 Torr of $H_2$ and 745 Torr of $O_2$ at room temperature. A catalyst sample was placed on a ceramic heater in the batch reactor, and the temperature was monitored by a thermocouple and fluctuated no more than 0.5 K. The reaction mixture was circulated continuously through the reaction line by a Metal Bellows recirculation pump at a rate of 2 L min$^{-1}$. The $H_2$ molecules were monitored as a function of reaction temperature (80−110 °C). After equilibrating for 1 h, the reaction mixture was continuously analyzed through an online gas chromatograph. The reaction mixture was separated for analysis using a DS iGC 7200 gas chromatograph equipped with a thermal conductivity detector and a 6 ft long, 1/8" outer diameter stainless steel 80/100 mesh size column. $H_2O$ conversion was reported in terms of TOF and was calculated on the basis of product molecules of $H_2O$ produced per metal surface site per second of reaction time.

**Characterization.** XPS spectra were acquired using a Thermo VG Scientific Sigma Probe spectrometer equipped with an Al–Kα X-ray source (1486.3 eV) and an energy resolution of 0.5 eV full width at half maximum under ultra-high vacuum conditions of $10^{-10}$ Torr. The TEM measurements were performed using a Tecnai TF30 ST at 300 kV and HRTEM, HAADF-STEM, and STEM-EDS mapping analyses were performed using a Titan cubed G2 60-300 (FEI) at 300 kV with a spherical aberration corrector (CEOS GmbH). EDS analysis was carried out along with four integrated silicon-drift EDS detectors (ChemiSTEM$^{TM}$ technology) at a collection solid angle of 0.7 srad. An aberration-corrected environmental TEM (Titan ETEM G2, FEI) operating at 300 kV with the Fusion heating holder (Protochips inc.) was used for the in situ TEM characterization. Field emission SEM was conducted using a Verios 460 SEM instrument. For electrical characterization of the nanodiodes, I–V curves and the short-circuit current were measured using a Keithley Instrumentation 2400 sourcemeter under various chemical reaction conditions. The reaction rates of the $H_2$ oxidation reaction were measured using a gas chromatograph (DS iGC 7200) in a batch reactor system.

**Simulation methods.** Spin-polarized density-functional theory calculations were performed to optimize the atomic structures and calculate the electronic energies using the Vienna Ab initio simulation package code[39–42]. The D3-corrected[43] revised Perdew–Burke–Ernzerhof exchange-correlation functional[44,45] with the projector augmented wave method was used[39,46], and the kinetic cutoff energy was set to 500 eV. The nudged elastic band method was employed to find a minimum energy pathway and to calculate the activation barriers[47,48]. Geometry optimization and nudged elastic band calculations were performed until the residual force on each atom was less than 0.05 eV Å$^{-1}$ and 0.1 eV Å$^{-1}$, respectively. K-points were sampled using a $2 \times 2 \times 1$ Monkhorst–Pack mesh[49]. To represent the Pt (111) surface, 48 atoms in a ($4 \times 4$) surface unit cell with three atomic layers was modeled with spacing more than 15 Å in the z-direction. To represent the CoO/Pt interface, we modeled periodic CoO islands on a three-layered Pt (111) slab (Supplementary Fig. 17)[29,35].

**Data availability.** The authors declare that all data supporting the findings of this study are available within the paper and its supplementary information files.

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

## Acknowledgements

This work was supported by the Institute for Basic Science (IBS) [IBS-R004]. K. An thanks the Basic Science Research Program through the National Research Foundation of Korea (NRF) funded by the Ministry of Education (2015R1C1A1A01055092). Y. Jung acknowledges support through the National Research Foundation of Korea from the Korean Government (NRF-2017R1A2B3010176, NRF-2016M3D1A1021147).

## Authors contributions

H.L. and J.Y.P. designed and the experiments. H.L. carried out the central measurements and analysis presented in this work. C.L. and H. L. fabricated the devices. K.A. and H.L. carried out the synthesis experiments. J.W.S., H.L., and R.R. performed the in situ TEM experiments. J.L., S.B., and Y.J. performed the DFT calculations. H.L., J.L., Y.J., and J.Y.P. wrote the manuscript and all authors edited the final manuscript.

## Additional information

**Competing interests:** The authors declare no competing interests.

