## [Peer Review File · Nature Communications]

Reviewers' comments:

Reviewer #1 (Remarks to the Author):

Lee and coworkers reported an interesting work on utilizing PtCo bimetallic nanoparticles for H₂ oxidation reaction. The authors provide pre- and post-XPS, XRD, TEM and STEM-EDX characterizations together with DFT calculations of OH formation at the interface of CoO island on a Pt(111) surface. Unfortunately, none of them is supporting the most important hypothesis in this study that CoO-Pt interface promotes electron transport and enhances the reactivity. I do not recommend acceptance of this manuscript in Nature Communication. My comments follow:

- 1) As mentioned by authors (line 131, page 6), Co NPs can be oxidized easily upon exposure to air, considering the challenge of operando XPS for HOR, authors should provide XPS data at working conditions or at least without exposure to air.
- 2) Except for the Co2p and Pt4f data presented in this paper, the authors should also provide and discuss the O1s and fermi spectra, although they might be influenced by TiO₂ and Au.
- 3) The authors should provide detailed information about the XPS fittings. In Supplementary Figure 8(a), the position of Co²⁺ 3/2p has about over 1eV shift among the three fresh samples, while in Supplementary Figure 10 the same fitted peak has over half eV shift, all of these are over the instrument resolution reported in the Methods section. The author should provide reasonable explanations; otherwise the compositional analysis reported in this paper is not right.
- 4) The microscope used by authors in this study should have a resolution roughly around 0.2 nm for C-term imaging. The 0.01 nm differences in Figure 2 (a-c) are not representative. Much higher resolution images should be provided to support the authors' conclusions. For example, a very recent paper reported by S. Dai et al. (Nature Communication, 8, 2017, 204) used aberration corrected microscope to investigate Pt-Co for ORR.
- 5) The XRD patterns reported in this paper only provide the formation of fresh Pt-Co bimetallic NPs. Again, it is not related to the formation of CoO-Pt interface or Co-O-Pt.
- 6) How do the authors know that the composition is uniform in each nanoparticle?
- 7) The DFT calculations were done on a slab of Pt with CoO on top of it, which is not in line with the structural information obtained by characterizations in this study. What is the structure of CoO overlayer on Pt? How does this affect the DFT results? The difference in OH barrier is too small to infer that one catalyst is better than the other (it is within error of DFT).
- 8) What evidence do the authors have that the chemistry doesn't happen on CoO clusters themselves?
- 9) The paper lacks comparison to other works on CoPt nanoparticles, e.g., by Schuth and co-workers (Nature Materials) and Gorte and co-workers (ACS Catalysis).

Reviewer #2 (Remarks to the Author):

The authors report an interesting study about the catalytic activity of bimetallic PtCo nanoparticles at metal oxide surfaces. Identifying the key properties defining catalytic activity of a surface is challenging because of the complexity of surface reactions: surface reactions involve a multitude of elementary steps and many degrees of freedom, nuclear and electronic, participate. Furthermore, the surface structure can change during the reaction, meaning that the structure of the catalytic active surface is not precisely known. For this reasons, it is very hard to model surface reactions and to predict the catalytic activity of a surface structure. One interesting group of systems are bimetallic nanoparticles deposited on a substrate. By changing the composition of the nanoparticles their properties can be tuned and catalytic activity can be optimized. The fundamental cause for the enhanced activity is still under discussion. The authors contribute to the discussion by studying the oxidation of H₂ on PtCo nanoparticles of different composition as a model system. They used catalytic nanodiodes to measure hot electrons created in the surface reaction, accompanied by XPS spectra of the nanoparticles before and after the reaction and relate the results to the catalytic activity of the nanoparticles by measuring turn over frequencies. The authors present elaborated experiments complemented by a theoretical model. The results support the hypothesis that the presence of oxide-metal interface sites leads to improved catalytic activity of the nanoparticles and gives valuable insights into the catalytic activity of bimetallic nanoparticles. The manuscript eventually merits for publication in Nature Communications. However, the following two issues should be addressed prior to publication.

One major challenge for experiments in this research area is to create well-defined and reproducible experimental conditions. The authors took great care in characterizing the catalytic nanodiodes used in the experiments and performed various tests to ensure that the observed current really originates from hot electrons created in the surface reaction. The tests are presented in the manuscript as well as in the extensive supplementary material and, in my opinion, provide convincing proof that the observed current is indeed chemi-current. The uncertainties in the measurements of the turn over frequency are well documented and discussed. The bimetallic nanoparticles are also well characterized in terms of size, composition and structure. However, one aspect, which is not discussed in the manuscript, is the influence of the underlying substrate. In case of the nanodiode, the nanoparticles are supported on a thin gold film on TiO₂ and in case of the turn over frequency measurements on SiO₂. Both substrates are very different and could affect the nanoparticles and their catalytic activity. This point is crucial here, because the authors calculated the chemi-current yield using the assumption that the reaction rate is equal in both cases. Based on this results (figure 3 e) the authors argue that a local polarization occurs due to charge transfer between metal and oxide (lines 223-230). The authors

should provide an estimate on how reliable these results really are.

In general, the authors should address the question, if there is an important effect correlated to electron hole pair excitation which influences the reaction mechanism. Do electron-hole-pair excitations really affect the reaction, or are they just an “irrelevant byproduct” that can be used to monitor the activity here? One of the key questions in chemical dynamics at surfaces is not, if electron hole pair excitations exist, but if they have a non-negligible effect on the reaction and if theories including electronic excitations are required to model reactions on surfaces. One way to address this question could be to compare the absolute magnitude of chemi-current observed in the presented experiments with the value reported by Nienhaus and coworkers for H adsorption on a metal surface. Following this line of thought, it should be discussed if an adiabatic model like the one used in the paper is sufficient here or if also non-adiabatic effects have to be considered to explain the enhanced reactivity of the PtCo nanoparticles.

Additionally, I have two comments the authors might consider in a revised version of the manuscript:

The key result of the presented work is that the origin of the synergistic effect on catalytic activity of the PtCo nanoparticles is related to the presence of oxide-metal interfacial sites, which form during the reaction on the surface. This conclusion is based on XPS spectra presented in the supplementary material, calculated chemi-current yields and is further supported by theoretical calculations. In my opinion the XPS spectra are very important for the discussion and I suggest to not only show the spectra in the supplementary material, but to add two exemplary XPS spectra (one before, one after reaction) to the manuscript.

The authors use the acronyms EDS and EDX in the manuscript which I suppose both stand for energy dispersed X-Ray spectroscopy. The authors should stick to one acronym for consistency.

Reviewer #3 (Remarks to the Author):

Park and coworkers reported the quantitative detection of hot electrons induced by H₂ oxidation on Pt-based nanoparticles by measuring the chemi-current on a catalytic nanodiode. They have observed the synergistic effect between Pt and Co, and the interface formation between CoO and Pt has been attributed to the enhanced catalytic performance. The strength of this work is that they have clearly shown the enhanced catalytic performance of Pt-catalyzed H₂ oxidation by the presence of Co. They could correlate the reaction rate or TOF with the chemi-current yield directly, such that the chemi-current provides a direct parameter to “visualize” the reaction performance. Furthermore, the role of CoO/Pt interface has been clearly revealed through such measurements. This is quite unique and of great importance. The weakness of the work is that the effect of the oxide/metal interfaces in catalytic reactions has been well understood. For example, the coverage of CoO_x overlayer on Pt-catalyzed surface reactions has been reported by Somorjai's group decades ago, who observed the maximum activity with a submonolayer oxide overlayers. Moreover, the effect of surfactants on bimetallic catalysts seems to make the

processes more complicated. Moreover, using nanoparticles the amount of surface Pt atoms can be varied with the ratio of Co/Pt, which is different with oxide-covered Pt single crystal surface (fixed amount of surface Pt atoms irrespective of the oxide coverage).

Reply to Reviewer 1's report

We thank the reviewer for their interest and valuable comments. Here, we would like to emphasize again the importance of the chemicurrent results of our work and provide additional *in situ* results, because of the comments regarding a lack of evidence supporting our conclusion, as mentioned below:

"...The authors provide pre- and post-XPS, XRD, TEM and STEM-EDX characterizations together with DFT calculations of OH formation at the interface of CoO island on a Pt(111) surface. Unfortunately, none of them is supporting the most important hypothesis in this study that CoO-Pt interface promotes electron transport and enhances the reactivity..."

The most important achievement in this work is visualizing the increased electron transport at the oxide–metal interface in bimetallic NPs through direct measurement of hot electrons. The highly enhanced chemicurrent yield in the Pt₃Co₁ NPs clearly demonstrates that hot electron transport is accelerated during the reaction because of the presence of the CoO-Pt interface, which is now confirmed by XPS analysis in this study. In fact, segregation of the CoO on the PtCo bimetallic NPs under oxidation conditions has been well verified and understood using *in situ* XPS and *in situ* TEM techniques. Recently, S. Dai et al. even clarified the fact that CoO is segregated more favorably on the {111} surface of Pt₃Co₁ NPs (~ 4 nm) while {100} surfaces resisted oxidation, which further implies the presence of the CoO-Pt interface on the Pt₃Co₁ NP surface (Nano Lett., 2017, 17, 4683–4688).

Along with this evidence, we decided to carry out *in situ* TEM experiments ourselves using the Pt₃Co₁ NPs we synthesized to provide strong support for our conclusion. We succeeded in observing atom-by-atom growth of CoO in real time on the surface of the Pt₃Co₁ NPs under oxidation conditions (see Fig. 4c and 4d). The results are consistent with the XPS results shown in this study as well as *in situ* TEM results in the previous studies. Based on the results from chemicurrent, TOF, XPS, and *in situ* TEM analyses and DFT calculations, it is obvious that a CoO layer is formed on the surface of the PtCo bimetallic NPs during the reactions; consequently, the CoO-Pt interface plays a crucial role in the enhanced catalytic reactivity and efficient charge transport. We added a detailed experimental method for and data from the *in situ* TEM measurements in the revised manuscript and supplementary information with a new Fig. 4 and Supplementary Figs.15 and 16. In response to all the of comments and criticisms raised, we believe that our claim of the formation of the Pt-CoO interface on the bimetal nanoparticles can be supported by these *in situ* TEM measurements. Detailed responses to the reviewer's other comments are given below.

Referee comment #1. *As mentioned by authors (line 131, page 6), Co NPs can be oxidized easily upon exposure to air, considering the challenge of operando XPS for HOR, authors should provide XPS data at working conditions or at least without exposure to air.*

Reply #1: Following the reviewer's comment, we conducted XPS analysis of the as-prepared Co NPs with minimal exposure to air because it is impossible to exclude air during the synthesis process. As shown in supplementary Fig. 12b, the Co 2p_{3/2} spectrum of the as-prepared Co NPs contains a peak at 781.1 eV of CoO (Co²⁺) with an intense shakeup peak and no metallic state is observed, implying that the Co is instantly oxidized, even with brief exposure to air, as reported in previous studies (S. Alayoglu et al., Top. Catal., 2011, 54, 778-785). The Co NPs are steadily more oxidized, therefore, more Co³⁺ states appear and transform to Co₃O₄ after 2–3 days of aging. The most important point here is that the monometallic Co NPs, which are readily oxidized as a form of CoO or Co₃O₄, did not show any catalytic activity for H₂ oxidation at temperatures of 80–110 °C.

Referee comment #2. *Except for the Co 2p and Pt 4f data presented in this paper, the authors should also provide and discuss the O1s and fermi spectra, although they might be influenced by TiO₂ and Au.*

Reply #2: As pointed out by the reviewer, in addition to the analysis for the Co 2p and Pt 4f XPS spectra, we characterized the corresponding O 1s and valance band (VB) spectra obtained from the Pt_xCo_y NPs. These results also support the presence of CoO on the surface of the PtCo bimetallic NPs.

In the XPS spectra attributed to O 1s, we could characterize three singlets at binding energies of 532.8, 531.5, or 529.8 eV, which correspond to Si–O, C=O, and Co oxide, respectively. As can be seen in Supplementary Fig. 13, the peaks at 532.8 and 531.5 eV show a strong intensity in the Pt₃Co₁ bimetallic NPs, whereas the peak signal associated with Co oxide (529.8 eV) is marginal. These results indicate that the O 1s peak for Co oxide formed on the NP surface was buried due to the relatively large signal from the C=O bond in the organic capping layers (PVP) as well as from the SiO₂ support (Si–O). This was further confirmed by the fact that the peak for Si–O become larger and the one for C=O become smaller after the H₂ reaction because of thermal decomposition of the capping layer. This tendency was the same for the other NPs (Pt₁Co₁ and Pt₁Co₃). However, in the case of Pt₀Co₁, where the NPs are twice as large as the other NPs (*i.e.* Pt₃Co₁, Pt₁Co₁, Pt₁Co₃ NPs) and the NPs are deposited on the SiO₂ substrate with higher coverage (~ 50 %), a relatively strong signal attributed to Co oxide (529.8 eV) was observed, implying that the metallic Co NPs are oxidized.

VB spectra of the Pt_xCo_y NPs were acquired to identify the electronic structure near the Fermi level and the oxidation states of Co depending on the surface composition. As shown in supplementary Fig. 14a, the VB spectra of the PtCo bimetallic NPs have a broad peak near the Fermi level, which is distinguished from the electronic structures of metallic Co

and Co_3O_4 . Because the broad features appear from multiple splitting of unpaired 3d electrons and p-d charge transfer satellite splitting of CoO, it is clear that CoO exists on the PtCo NPs. We also observed that the density of states near the Fermi level decreased slightly with increasing Co, which implies that the higher the Co content, the greater the CoO, which is in agreement with the Co 2p spectra. This tendency was confirmed more clearly by a rapid decrease in the electronic density of the PtCo bimetallic NPs after H_2 oxidation (Supplementary Fig. 14 b–f). A detailed description of the data is added in the revised manuscript and supplementary information (Supplementary Figs.13 and 14). While we find that additional analysis of the O 1s and VB spectra is important, we believe that the formation of Pt-CoO can be supported by the *in situ* TEM measurements carried out separately.

Referee comment #3. *The authors should provide detailed information about the XPS fittings. In supplementary Figure 8(a), the position of Co^{2+} 3/2p has about over 1 eV shift among the three fresh samples, while in Supplementary Figure 10 the same fitted peak has over half eV shift, all of these are over the instrument resolution reported in the Methods section. The author should provide reasonable explanations; otherwise the compositional analysis reported in this paper is not right.*

Reply #3: We thank the reviewer for this meticulous comment regarding XPS fitting. Accordingly, we checked all of the XPS data again and recognized that we made a mistake in fitting the Co 3/2p peak in supplementary Fig. 8a, which is now Fig. 10a in the revised manuscript. In the previous analysis, the fitting considered that both the Co^{3+} and Co^{2+} species coexisted as oxidation states of cobalt, however, we overlooked the fact that the position of the Co^{2+} peak was shifted by 1 eV. However, the fitting is still reasonable when we only take into account the Co^{2+} oxidation state (at ~ 781.1 eV) with a strong satellite peak (at ~ 786.0 eV), as shown in the revised supplementary Fig. 10a. Therefore, we could confirm that the oxidation state of the cobalt oxide in the Pt_1Co_3 NP was Co^{2+} , representing CoO, as in the other PtCo bimetallic NPs. The compositional analysis with the newly fitted data also showed the same atomic ratio as the previous result.

However, in fitting the Co 3/2p peak in supplementary Fig. 12b, the spectra were correctly deconvoluted into two main peaks corresponding to Co^{3+} at ~ 779.8 eV, Co^{2+} at ~ 781.1 eV with two small satellite peaks of Co^{3+} and Co^{2+} states (at ~ 785.4 eV and ~ 789.3 eV, respectively), indicating that the pure Co had been oxidized to Co_3O_4 . Therefore, we only modified the figure and now include clearly marked oxidation states to avoid confusion.

Speaking of detailed information about the XPS fitting, to ensure accurate analysis, all the XPS spectra were calibrated on the basis of the adventitious C 1s peak at 284.8 eV of binding energy and the inelastic backgrounds of all the spectra were subtracted using the Shirley background. For charge compensation at the surface, a flood gun was used during all scanning processes. For quantitative analysis of the composition, the peak area of each element (*i.e.* Co 2p, Pt 4f) was normalized with XPS sensitivity factors (12.6 for Co 2p and 15.5 for Pt 4f) and the atomic ratio for all Pt_xCo_y NPs were estimated (Fig. 2d,

Supplementary Fig. 3, and Table 1). These points are all included in the revised manuscript and supplementary information.

Referee comment #4. *The microscope used by authors in this study should have a resolution roughly around 0.2 nm for C-term imaging. The 0.01 nm differences in Figure 2 (a-c) are not representative. Much higher resolution images should be provided to support the authors' conclusions. For example, a very recent paper reported by S. Dai et al. (Nature Communication, 8, 2017, 204) used aberration corrected microscope to investigate Pt-Co for ORR.*

Reply #4: We thank the reviewer for pointing this out. In fact, we acquired TEM images using both Tecnai TF30 ST (FEI) and Titan cubed G2 60-300 (FEI) systems. In particular, all the HRTEM images shown in Fig. 2 were obtained from the Titan cubed G2 60-300 (FEI), which has the proper specifications for high resolution (information limit of 80 pm) as the TEM (JEM-3100F, JEOL) mentioned by the reviewer (Nature Communication, 8, 2017, 204). However, we missed including it in the manuscript and only mentioned the Tecnai TF30 ST (FEI). Therefore, we performed the precise characterization of *d*-spacing for the NPs again and edited the manuscript, specifically the methods section and the supplementary information.

Referee comment #5. *The XRD patterns reported in this paper only provide the formation of fresh Pt-Co bimetallic NPs. Again, it is not related to the formation of CoO-Pt interface or Co-O-Pt.*

Reply #5: As for the XRD patterns, they were measured to provide the information that the as-synthesized PtCo bimetallic NPs form an alloy such that as more Co was incorporated into the Pt fcc structure to form an alloy, the peak position was shifted to higher angles. Therefore, as the reviewer mentioned, it is true that the XRD patterns reported in this paper are not related to the formation of the CoO-Pt interface. To confirm the formation of CoO on the PtCo bimetallic NPs during the reaction, sophisticated analysis of XPS and *in situ* TEM were conducted after and during the reactions, respectively, and all the results clearly demonstrated that the cobalt was easily oxidized and segregated on the surface of the PtCo NPs as a form of CoO under the reaction conditions. A detailed explanation of this point is given in the revised manuscript.

Referee comment #6. *How do the authors know that the composition is uniform in each nanoparticle?*

Reply #6: The elemental distribution of the Pt and Co within the PtCo bimetallic NPs was investigated by EDS mapping in STEM mode, as shown in supplementary Fig. 2. The results of the STEM-EDS mapping indicate that both the Pt and Co atoms are distributed

homogeneously in the alloyed PtCo NPs and that the atomic ratio of Co in these NPs (obtained from an area scan analysis) are about 22.7 %, 48.9 %, and 83.9 % for Pt₃Co₁, Pt₁Co₁, and Pt₁Co₃ NPs, respectively, which is comparable to our XPS and ICP-MS results shown in Fig. 2d. Furthermore, to obtain definite proof of the compositional uniformity of each NP, we conducted additional EDS mappings on many other NPs and clearly proved that all the individual NPs contain both Pt and Co with a constant ratio (Supplementary Fig. 3a). The homogeneous atomic distribution of the bimetallic NPs was also confirmed by area scan analysis in different regions of the NPs (Supplementary Fig. 3b). All the chemical mapping with STEM-EDS was carried out in the TEM (Titan cubed G2 60-300, FEI) at 300 kV along with four integrated silicon-drift EDS detectors (ChemiSTEMTM technology) at a collection solid angle of 0.7 srad. Additional data and the experimental details are added in the revised manuscript and supplementary information.

Referee comment #7. *The DFT calculations were done on a slab of Pt with CoO on top of it, which is not in line with the structural information obtained by characterizations in this study. What is the structure of CoO overlayer on Pt? How does this affect the DFT results? The difference in OH barrier is too small to infer that one catalyst is better than the other (it is within error of DFT).*

Reply #7: Through further *in situ* TEM experiments, we found that cobalt oxide is present on the surface of the platinum nanoparticles. Based on these data, we proposed the island model of a cobalt oxide cluster on a platinum surface and investigated the H₂ oxidation reaction therein. The island model for the CoO/Pt used for the DFT calculations (shown in Fig 5 and supplementary Fig. 17) was adopted from previous experimental and theoretical studies on the FeO island model on Pt (FeO/Pt: ref 29) and the CoO island on Pt (CoO/Pt: ref 35). It is true that the absolute errors of conventional DFT (GGA) can be comparable to 0.1–0.2 eV, but considering the internal consistency of the DFT when using the same functional, basis, and consistent models for comparison (which is the reason why DFT is so widely used despite the cited absolute error ranges), we believe the relative difference of 0.16 eV obtained for the CoO/Pt vs. Pt under the same methods indeed suggests a sizable qualitative difference, which can reflect reasonably well an experimental difference between the CoO/Pt and Pt.

Referee comment #8. *What evidence do the authors have that the chemistry doesn't happen on CoO clusters themselves?*

Reply #8: Following the reviewer's comment, to assess possible reactions on the CoO cluster, H₂ oxidation was further examined at the top site of the island model. As shown in supplementary Fig. 22, a large activation energy (1.34 eV) is required for the last step of H₂ oxidation on the top sites of a CoO cluster (compared to 0.72 eV for the rate-determining barrier at the CoO/Pt interfacial sites), so we can confirm that it would be difficult for the reaction to occur on the CoO clusters.

Referee comment #9. *The paper lacks comparison to other works on CoPt nanoparticles, e.g., by Schuth and co-workers (Nature Materials) and Gorte and co-workers (ACS Catalysis).*

Reply #9: As suggested by the reviewer, we now compare the structural characteristics of the PtCo bimetallic NPs based on the information reported in these previous studies. We added the following recommended citations in the manuscript:

- 1) G. Wang et al., Nature Mater., 2014, 13, 293-300
- 2) J. Lio et al., ACS Catal., 2016, 4, 4095-4104

Reply to Reviewer 2's report

We thank the reviewer for the acknowledgement of the importance of this study and favorable recommendation for publication with relatively minor revisions. We particularly appreciate this high evaluation, *“The authors present elaborated experiments complemented by a theoretical model. The results support the hypothesis that the presence of oxide-metal interface sites leads to improved catalytic activity of the nanoparticles and gives valuable insights into the catalytic activity of bimetallic nanoparticles. The manuscript eventually merits for publication in Nature Communications...”*. To address the reviewer's comments, we carried out additional experiments and added some discussion. Detailed responses to the comments are given below.

Referee comment #1. *The tests are presented in the manuscript as well as in the extensive supplementary material and, in my opinion, provide convincing proof that the observed current is indeed chemicurrent. The uncertainties in the measurements of the turn over frequency are well documented and discussed. The bimetallic nanoparticles are also well characterized in terms of size, composition and structure. However, one aspect, which is not discussed in the manuscript, is the influence of the underlying substrate. In case of the nanodiode, the nanoparticles are supported on a thin gold film on TiO₂ and in case of the turn over frequency measurements on SiO₂. Both substrates are very different and could affect the nanoparticles and their catalytic activity. This point is crucial here, because the authors calculated the chemicurrent yield using the assumption that the reaction rate is equal in both cases. Based on this results (figure 3 e) the authors argue that a local*

polarization occurs due to charge transfer between metal and oxide (lines 223-230). The authors should provide an estimate on how reliable this results really are.

Reply #1: We thank the reviewer for this compliment and comment. To clarify the influence of the substrate on the TOF results, we conducted additional TOF measurement on the Pt₃Co₁ NPs supported on a 10 nm Au film on SiO₂ under identical reaction conditions. As shown in supplementary Fig. 8, the catalytic activity of the Pt₃Co₁ NPs supported on the 10 nm Au film is comparable to the results of the Pt₃Co₁ NPs on SiO₂, where the difference is in the range of the measurement error (< 10 %). These experimental results reveal that the effect of the underlying substrate on the catalytic activity is negligible, thus the values of chemicurrent yield shown in this study are indeed reliable. We include this point in the revised manuscript and supplementary information.

Referee comment #2. *In general, the authors should address the question, if there is an important effect correlated to electron hole pair excitation which influences the reaction mechanism. Do electron-hole-pair excitations really affect the reaction, or are they just an “irrelevant byproduct” that can be used to monitor the activity here? One of the key question in chemical dynamics at surfaces is not, if electron hole pair excitations exist, but if they have a non-negligible effect on the reaction and if theories including electronic excitations are required to model reactions on surfaces. One way to address this question could be to compare the absolute magnitude of chemicurrent observed in the presented experiments with the value reported by Nienhaus and coworkers for H adsorption on a metal surface. Following this line of thought, it should be discussed if an adiabatic model like the one used in the paper is sufficient here or if also non adiabatic effects have to be considered to explain the enhanced reactivity of the PtCo nanoparticles.*

Reply #2: As the reviewer pointed out, we absolutely agree that it is a very important issue to know whether electronic excitation on the catalysts directly affect the processes of surface reactions. In this study, however, we focused on monitoring the hot electrons, which are evidence of non-adiabatic energy transfer in surface reaction processes, and addressing the enhanced catalytic activity at the oxide/metal interface on the PtCo bimetallic NPs. The present results have a huge significance because the unique electronic characteristics of the oxide–metal interface in bimetallic NPs have been visualized by detecting hot electrons in real time under atmospheric reaction conditions. With both the chemicurrent and reaction rates, the detection probability of hot electrons (*i.e.* so-called chemicurrent yield; the number of electrons detected per molecule of product) was estimated and the values have been reported in the range of 10⁻⁶–10⁻³ in previous studies. The values calculated in the present study (~ 10⁻⁵) is lower than the values obtained in Nienhaus’ works (10⁻⁴–10⁻³) because there is an additional potential barrier at the interface between the NPs and the Au thin film in our system, which disturbs the hot electron flow across the device. However, the comparison of chemicurrent yield does not give us a clear answer to the question of how much energy is

transferred via the non-adiabatic transition and how the generated hot electrons affect surface reactions. Here, the main message of our work is elucidating the existence of hot electrons on the complex bimetallic NPs during the surface reaction and enhanced electronic transport at the nanoscale oxide/metal interface on the NPs. Therefore, we think that a quantitative description regarding the role of hot electrons in the surface reaction is beyond the scope of this study. To further understand this process, we need a totally different experimental setup where vibrational or translational relaxation of the reactants can be observed, as reported in a previous study (O. Bünermann et al., Science, 2015, 350, 6266, 1346-1349).

Referee comment #3. *In my opinion the XPS spectra are very important for the discussion and I suggest to not only show the spectra in the supplementary material, but to add two exemplary XPS spectra (one before, one after reaction) to the manuscript.*

Reply #3: Following the reviewer's suggestion, we decided to include the two Co 2p XPS spectra for Pt₃Co₁ NPs obtained before and after the reaction to emphasize the formation of CoO on the surface of NPs during the H₂ oxidation reaction (Fig. 4a and 4b). In addition, we also present *in situ* TEM results (Fig. 4c and 4d) demonstrating the structural configuration of the CoO layer on the NPs, thus allowing the manuscript to clearly deliver the main message of our work.

Referee comment #4. *The authors use the acronyms EDS and EDX in the manuscript which I suppose both stand for energy dispersed X-Ray spectroscopy. The authors should stick to one acronym for consistency.*

Reply #4: As suggested by the reviewer, we now use only one acronym (EDS) for energy X-ray spectroscopy for clarity.

Reply to Reviewer 3's report

We thank the reviewer for the acknowledgement of the importance of this study and the favorable recommendation for publication with relatively minor revisions. We particularly appreciate this high evaluation, “ *The strength of this work is that they have clearly shown the enhanced catalytic performance of Pt-catalyzed H₂ oxidation by the presence of Co. They could correlate the reaction rate or TOF with the chemiurrent yield directly, such that the chemiurrent provides a direct parameter to “visualize” the reaction performance. Furthermore, the role of CoO/Pt interface has been clearly revealed through such*

measurements. This is quite unique and of great importance....”. Regarding the point raised by the reviewer as a weakness of the work, we would like to emphasize again the distinction of this work in demonstrating the role of the oxide/metal interface located in the bimetallic NPs through direct detection of hot electrons. Detailed responses to the comments are given below.

Referee comment #1. *The weakness of the work is that the effect of the oxide/metal interfaces in catalytic reactions has been well understood. For example, the coverage of CoO_x overlayer on Pt-catalyzed surface reactions has been reported by Somorjai’s group decades ago, who observed the maximum activity with a submonolayer oxide overlayers.*

Reply #1: As far as we know, no studies have been reported on the CoO_x overlayer on Pt-catalyzed surface reactions, although there have been some studies conducted on other oxide/metal systems such as AlO_x/Rh, TiO_x/Rh, VO_x/Rh, FeO_x/Rh, ZrO_x/Rh, NbO_x/Rh, TaO_x/Rh, and WO_x/Rh (A. Boffa, C. Lin, A. T. Bell, G. A. Somorjai, Promotion of CO and CO₂ Hydrogenation over Rh by Metal Oxides: The Influence of Oxide Lewis Acidity and Reducibility, *J. Catal.*, 1994, 149, 149–158). According to these studies, the rate of CO and CO₂ hydrogenation was enhanced when the monolayer oxide coverage on the Rh metal foil is about 0.5 monolayer and the maximum enhancement varied dependent on the type of oxide. These results suggested that the main active sites are at the oxide/metal interface sites, which is on the same conclusion with our work. However, it must be clearly distinguished that the significance of our study lies in verifying the presence of the oxide/metal (*i.e.* CoO/Pt) interfaces on the PtCo bimetallic NPs and demonstrating the unique electronic properties of the nanoscale oxide/metal interface through hot electron detection. Therefore, we believe that our results have a broad impact and interest to the heterogeneous catalysis and surface chemistry communities.

Referee comment #2. *Moreover, the effect of surfactants on bimetallic catalysts seems to make the processes more complicated. Moreover, using nanoparticles the amount of surface Pt atoms can be varied with the ratio of Co/Pt, which is different with oxide-covered Pt single crystal surface (fixed amount of surface Pt atoms irrespective of the oxide coverage).*

Reply #2: We thank the reviewer for noting these points. We agree that the surfactants used for stabilizing the NPs against aggregation are an important parameter affecting the catalytic activity. In this study, however, although the composition of the PtCo bimetallic NPs changed, the type and concentration of the surfactant (*i.e.* PVP) used to synthesize the NPs was the same. Therefore, it can be assumed that the amount of surfactant on the surface of all the NPs is comparable. In addition, according to the O1s XPS spectra analysis (Supplementary Fig. 13), it is obvious that a large amount of surfactant existing on the surface of the PtCo bimetallic NPs was thermally decomposed during H₂ oxidation and that any surfactant

residue on the PtCo bimetallic NPs (*i.e.* Pt₃Co₁, Pt₁Co₁, Pt₁Co₃ NPs) is insignificant. Based on this, we believe that the effect of the surfactants is negligible in this study.

As for the second point on the amount of surface Pt atoms with different ratios of Co/Pt, it is true that accurate control of the amount of surface Pt atoms is challenging in the PtCo bimetallic NPs despite careful regulation of the composition. However, when the Co is segregated and oxidized as a form of CoO on the surface of the NPs, the metallic Pt remains on the surface; therefore, it is presumed that most of the CoO layer is surrounded by a Pt-rich surface, as reported in a previous study (H. L. Xin et al., *Nano Lett.*, 2014, 14, 3203-3207). This indicates that the interfacial area of the CoO/Pt decreased with increasing ratios of Co/Pt, even though we could not systematically control the interfacial area. In this study, therefore, we would like to focus more on the formation of the oxide/metal interface on the bimetallic NPs under reaction conditions, which was confirmed by convincing evidence observed via XPS, *in situ* TEM, and chemiurrent measurements, and the improved catalytic activity of the catalysts at this oxide/metal interface. As a subsequent study, a new design of nanocatalysts can be suggested, where a differing amount of CoO is attached on Pt NPs synthesized in advance to fix the amount of surface Pt atoms irrespective of oxide coverage. We believe that this can provide a profound insight regarding the effect of the oxide/metal interface at nanoscale.

Reviewers' Comments:

Reviewer #1 (Remarks to the Author):

The authors have adequately revised the manuscript and is now suitable for publication.

Reviewer #2 (Remarks to the Author):

Extensive experimental work was added to the manuscript to address to issues brought up by the referees. The additional data improves the quality of the manuscript and give an even more detailed picture of the reaction. In particular, experiments were performed to check the effect of the substrate on the catalytic activity, which turns out to be negligible. Like I already stated in the first report, I believe the paper presents very interesting experiments supported by theoretical calculations giving new insight into the catalytic activity of bimetallic nanoparticles. In my opinion the paper in its current form is suited to be published in nature communications.

Reviewer #3 (Remarks to the Author):

I recommend the revised manuscript for publication in NC.